# Coordinating principal–agent and incentive strategy of cold chain logistics service in fresh food supply chain

**Ming Zeng, Yuxiang Wu****\*, Xiaoling Xing, Wenjing Tang, Huyang Xu**

College of Management Science, Chengdu University of Technology, Chengdu, People's Republic of China

\* m13098776015@163.com

## Abstract

With the increasing production and circulation of fresh food, society's requirements for product quality have also increased. Currently, upstream and downstream enterprises in the fresh food supply chain tend to delegate the cold chain transportation of fresh food to third–party cold chain logistics (CCL) service providers who offer more professional integrated preservation services. This study adopts coordination theory to research the delegation, coordination, and incentive strategies between a Fresh Food Producer, Distributor, and CCL Service Provider. The aim is to promote the efforts of the CCL Service Provider in improving freshness and achieving the overall optimal interests of the fresh food supply chain. First, the basic models of the Fresh Food Producer and Distributor individually entrusting the CCL Service Provider are established under different information conditions. Second, a collaborative delegation agency model between the Fresh Food Producer and Distributor is established on top of the basic models. Finally, through the optimal decision solutions under different information conditions and numerical calculations of preservation benefit levels, the following conclusions are drawn: (1) The more transparent and open the information environment, the more it can encourage the CCL Service Provider to improve the quality of their preservation services, allowing the Fresh Food Producer and Distributor to obtain more significant preservation benefits. However, when information is completely transparent, the incentive mechanisms formulated by the Fresh Food Producer and Distributor become ineffective. (2) Collaborative cooperation strategies can encourage the CCL Service Provider to enhance their preservation efforts while strengthening the effectiveness of relevant incentive strategies for the Fresh Food Producer and Distributor.

## 1. Introduction

According to the official website of the National Bureau of Statistics of China, the total grain production in China exceeded 2 billion tons in 2022, with the production of fresh food surpassing 1.2 billion tons [1, 2]. Fresh food is different from ordinary products in that they are perishable and susceptible to deterioration, so they have high requirements for the delivery timeframe and freshness preservation environment. Compared with the transportation loss

**Data Availability Statement:** All relevant data are within the manuscript and its Supporting Information files.

**Funding:** Funder: National Natural Science Foundation of China (Grant No. 72101036) HYX

300,000¥ Huyang Xu https://kd.nsfc.cn/fundingProjectInit The funder played role in the research process of the manuscript. Funder: Humanities and Social Sciences Research Fund of Ministry of Education HYX (Grant No. 21YJC630145) 80,000¥ Huyang Xu http://www.moe.gov.cn/s78/A13/tongzhi/202108/t20210820_552619.html The funder played role in the data collection process of the manuscript. Funder: Philosophy and Social Science Research Fund of Chengdu University of Technology (Grant No. YJ2022-YB021) MZ 10,000¥ Ming Zeng https://skc.cdut.edu.cn/info/1127/2848.htm The funder played role in computational experiments of the manuscript.

**Competing interests:** The authors have declared that no competing interests exist.

rate of fresh food in developed countries, the transportation loss rate of fresh food in China is very high. The loss rate of fresh vegetables and fruits during the "first kilometer" of transportation is as high as 15%–25%, resulting in an annual loss of 12 million tons of fruits and 130 million tons of vegetables. By contrast, developed countries with high–quality logistics and transportation infrastructure, only have a 5% the transportation loss rate of fresh produce [3].

The traditional wholesale market model is the most common in China's fresh food distribution and transportation system, where fresh food is distributed layer by layer through distributors and retailers before reaching consumers. However, the transportation links of this supply chain are redundant, and the loss rate of fresh food is high. Self–built cold chain logistics (CCL) is the future development trend of fresh food logistics and distribution. Under the control of cold chain equipment, the freshness of fresh food is guaranteed, effectively controlling perishable losses. However, building an in–house CCL system is costly and difficult for the majority of fresh food companies to afford. As a result, most fresh food companies outsource their non–core business to specialized CCL Service Providers, integrating their resources to gain greater market competitiveness. By relying on a CCL Service Provider for distribution, both participants in the upstream and downstream participants in the fresh food supply chain can effectively reduce operating costs, minimize unnecessary losses in circulation, and ensure that fresh produce reaches the market in optimal condition [4]. Researchers have conducted some preliminary explorations on the principal–agent model of entrusting the cold chain to a third–party CCL Service Provider in the upstream and downstream enterprises of existing fresh produce supply chains. Some have explored the cold chain principal–agent scenarios under different power structures, such as producer–led or retailer–led [5, 6], while others have considered the integrity of the supply chain by integrating the fresh produce transportation tasks of upstream and downstream enterprises in the supply chain [7], entrusting them to the same CCL Service Provider agent. Information asymmetry will arise between the CCL Service Provider and the entrusted party, Thus, to maximize operating profits, the CCL Service Provider tends to selectively withhold useful information from the other party. At the same time, it is more likely that the CCL Service Provider will have insufficient preservation efforts, as their overall interests of in the fresh food supply chain are low.

Above all, this study will utilize synergy theory to conduct in–depth research on the principal–agent coordination and the incentive strategy issues between Fresh Food Producers, Distributors and CCL Service Providers. The following research questions (RQs) will be addressed:

1.How do information conditions affect the coordination strategies and incentive mechanisms of fresh food supply chains? Which information conditions are favorable for the upstream and downstream of the fresh food supply chain to obtain greater preservation benefits?

2.What are the incentive effects of coordination strategies upstream and downstream of the fresh food supply chain and how do they affect the level of effort of the CCL Service Provider?

3.How does the coordination strategy affect the upstream and downstream of the fresh food supply chain?

Regarding RQ 1, it can be concluded that under conditions of complete information, the preservation incentive strategies of fresh agricultural producers and distributors are ineffective. The incentive coefficient has no correlation with factors such as effort output, and effort cost coefficient. Under conditions of incomplete information, the incentive coefficient is positively correlated with effort output coefficient, and negatively correlated with effort cost coefficient and risk aversion coefficient. Regarding RQ 2, it can be concluded that in the supply chain of fresh agricultural products, the preservation benefits of producers and distributors are negatively correlated with the effort level and effort cost of CCL service providers, and positively

correlated with effort output. This result implies that CCL service providers are more able to increase their effort level when taking on tasks with lower effort costs and higher effort output, and can simultaneously enhance the preservation benefits of fresh food producers and distributors. Regarding RQ 3, it can be concluded that when fresh food producers and distributors cooperate in a coordinated manner, under equal information conditions, the preservation effort level of CCL service providers and the incentive coefficients of fresh food producers and distributors are higher than in the base model. From the perspective of supply chain coordination strategy selection for fresh food producers and distributors, adopting a collaborative cooperation supply chain coordination strategy does not immediately result in preservation benefits. Nonetheless, with increasingly close information exchange, the long–term preservation benefits of the fresh agricultural supply chain can be significantly improved.

The rest of this paper is organized as follows. Section 2 provides a literature review, and Section 3 illustrates three principal–agent models. Sections 4 and 5, provide comparative and sensitivity analyses, respectively. Section 6 concludes the paper. To improve the readability of the paper, we also present a detailed proof procedure in the S1 Appendix.

The contributions of this study are as follows:

(1) From the theoretical perspective.

This study discusses the problem of coordinating strategies of upstream and downstream enterprises in CCL service entrusted agency from the supply chain perspective and constructs an entrusted agency model of cold chain service for fresh agricultural products.

In the selection of supply chain coordination and incentive strategies, the synergy theory is incorporated into the principal–agent framework of cold chain services in fresh produce supply chains, and the interaction between supply chain strategies and the principal–agent framework is explored.

(2) For the managerial perspective.

With established systems in place, it will be easier for the fresh food supply chain cold chain service principal–agent participants to formulate reasonable incentive and preservation strategies as well as strengthen the information sharing and synergistic assistance between upstream and downstream enterprises in the fresh food supply chain. Consequently, the level of preservation efforts will be improved, and high–quality fresh agricultural products can be provided for consumers.

## 2. Literature review

This study covers several areas of research. In the following, we briefly review some relevant prior studies and show the literature orientation of this paper.

### 2.1 Fresh food supply chain and its cold chain transportation

Current research in the fresh food supply chain includes inventory management [8–10], supply chain design [11, 12], and product pricing strategies [13], Bo Yan et al. examined the impact of different transportation modes on supply chain performance and analyzed the operational strategies of the fresh food supply chain under three scenarios [14]. Rahdar et al. designed a coordination mechanism for deteriorating products in a two–level supply chain system and assumed that both the demand rate and deterioration rate of inventory products are deterministic [15].The above studies indicate that research on fresh food is focused primarily on product loss, which typically involves two types: quantity and quality loss [16]. Liu et al. conducted research on pricing and preservation decisions in a dual–channel supply chain for fresh food under demand information uncertainty. Their study revealed that inventory and shortage costs can reduce the optimal pricing and preservation efforts of the manufacturer

[17]. Liu et al. used an exponential function to explore the relationship between the quantity loss of fresh agricultural products and time, and offline producers model dynamic freshness preservation efforts [18]. Zhang et al. considered the effect of investment level on quantity loss and price–dependent demand for spoiled products and developed a single–manufacturer, single–buyer supply chain model [19]. The regional, seasonal, perishable, and expendable characteristics of fresh agricultural products make their transportation mode different from other types of products. Many scholars are committed to optimizing the logistics and transportation network for fresh food, so that it is delivered faster and fresher to consumers [20]. De Keizer et al. developed a network design model to improve the performance of logistics networks [21]. Blackburn and Scudder designed a hybrid response model to minimize the value loss of fresh food [11]. Based on the above literature, most studies on the fresh food supply chain and its cold chain transportation are focused on inventory optimization, pricing and planning, and transportation optimization. Less attention has been paid to the upstream and downstream enterprises of fresh food supply chain entrusted to third–party CCL service enterprises.

## 2.2 Cold chain logistics service commissioning agent

For fresh food companies with a strong demand for CCL, entrusting the task of fresh food transportation to a suitable cold chain service company is a more cost–effective and widely used approach. Zhang et al. introduced the concept of the Stackelberg game and analyzed the relationship between fresh suppliers and CCL Service Provider [22]. Feng, Singh et al. studied the transportation of perishable products in an uncertain environment by using third–party logistics services as a mixture flow operation [23]. Zhang et al. studied the impact of digital transformation based on blockchain technology under cold chain conditions on the delegation and agency of manufacturers, retailers, and third–party logistics service providers [24]. Huanga et al. explored the third–party logistics principal–agent problem from a risk management perspective to motivate third–party logistics service providers to provide higher levels of effort [25]. Yu et al. considered the effect of the CCL Service Provider's effort level on the quantity and quality loss of fresh food. They investigated the optimal decision of pricing fresh food and the CCL Service Provider's effort level under supplier–driven and cold chain service provider–driven scenarios [7]. Hofer et al. found that in the CCL principal–agent relationship, the justice dimension has an important influence, while the strategic fairness of the CCL Service Provider has an important role in fostering the customers' level of trust [26]. In addition, some scholars found that information is another important factor affecting the principal–agent relationship of CCL services. Zhou et al. designed a contractual coordination mechanism that can solve the production information asymmetry of fresh food and provide a coordination solution for supply chains with uncertain demand and unclear product channel information [27]. Ketzenberg et al. investigated the inventory management problem for perishable products and discussed the value of information sharing [28]. Liu et al. examined that incentive strategies and information sharing in the supply chain of a fresh food e–commerce company, where fresh food suppliers and fresh food e–commerce companies are each responsible for the preservation tasks and value–added services, respectively [29]. Fu Jia developed a coordination mechanism based on a dual–form game, combining non–cooperative and cooperative games to coordinate a multi–agent reverse supply chain with one remanufacturer and two competing collectors. The coordination problem of a multi–agent reverse supply chain with collector fairness considerations was studied [30]. Most of the current studies on fresh food and CCL service principal–agent are from the perspectives of risk management, power structure, and information transparency to study the impact of these factors on principal–agent dynamic.

However, there are a few studies on the coordination strategies of upstream and downstream firms in CCL service principal agency from the supply chain perspective.

## 2.3 Supply chain coordination and incentive strategies

Coordination in supply chains is usually achieved through contracts between upstream suppliers and downstream fresh food distributor to increase the total profit of the supply chain [31]. Some feasible coordination incentive strategies have been proposed for supply chain coordination, such as buybacks [32], quantity discounts [33], revenue sharing [34], and price discounts [33], but most of these studies are only for two–level supply chains. For the supply chain coordination problem with multiple participants, Mirzapour et al. studied a three–level supply chain consisting of multiple suppliers, multiple manufacturers, and multiple customers, with uncertainty in both cost parameters and demand [35]. Zhang and Liu designed a revenue–sharing mechanism and other coordination mechanisms for a three–level (supplier–manufacturer–retailer) green supply chain under a noncooperative game [36]. Zhang utilized the Stackelberg game method to solve the optimal pricing and optimal preservation decisions for Fresh Food Producer and Distributors under centralized and decentralized decision–making models [37]. Perishable product deterioration rates are time–sensitive in the sense that fresh food deteriorates at a constant rate. For this reason, Panda et al. addressed the problem of coordination and profit sharing in a three–tier supply chain consisting of manufacturers supplying perishable products to retailers in a single batch through a Fresh Food Distributor [38]. Chongfeng Lan introduced promotional efforts and service levels into system decision–making. To achieve the collaborative operation of Guangfa Bank's supply chain, a revenue–sharing commission coordination contract was designed using the coordination tool entrusted by group leaders, combined with traditional revenue–sharing contracts. The relevant theoretical basis provides reference for enterprise decision–making [39]. Nana Wan et al provided explicit option coordination conditions for disrupted supply chains under two supply chain structures, and explored the effects of interruptions and supply chain structures on option coordination conditions. They compared the high and low profit distribution coefficients on option prices and exercise prices dominated by different themes [40]. Long Guo et al. studied how to utilize the third-party contract to coordinate the revenue and profit distribution between the two, and the proposed flexible ordering strategy significantly improves the overall revenue level of the logistics service supply chain by coordinating the contract [41]. Based on the coupled coordination theory, Tianqi Peng analyzed the coordinated development of the regional economy and ecological environment of the logistics industry in Anhui Province, and analyzed the factors restricting the coordinated development of the three systems [42]. Umer Mukhtar et al. contributed to the literature related to coordination of reverse supply chain systems by identifying equilibrium solutions for selling, acquisition and transfer prices of remanufactured products in decentralized, centralized and coordinated environments [43]. Regarding the literature on coordination theory in terms of fresh cold chain supply chain service principal-agent, in which the literature considering the degree of information completeness is still missing, and the literature in terms of fresh cold chain supply chain cooperation considering the comparative aspect of the degree of information completeness, it mostly discusses pricing, profit distribution, and does not consider the logistic service principal-agent coordination and incentive strategies.

In recent years, some scholars have studied supply chain coordination from the perspective of collaborative cooperation among supply chain participants. Wike et al. argued that collaborative cooperation enables each participant to share their capital and technology, which reduces uncertainty in task completion, achieves shared costs and direction among all parties,

and increases supply accuracy [44]. Zhou et al. studied the problem of retailer–dominated supply chain alliance selection [45]. Glock and Kim discussed the impact of collaborative cooperation on operational policies by considering three types of competition among retailers [46]. Guoxuan Huang et al explored the supply chain coordination and agency problem of charitable donations through Stackelberg game theory. E–commerce platforms determine product sales and commission, while manufacturers determine charitable donations, investment, and price, as well as study charitable donations and pricing strategies in centralized and decentralized systems [47]. Yi Wan et al. established a green supply chain model that includes an e–commerce platform and a manufacturer. They used Nash and Rubinstein bargaining contracts to adjust platform usage rates for coordination, alleviate profit conflicts, promote green product development, and achieve a win–win situation [48]. Therefore, based on the above research ideas, the present study will consider the supply chain coordination strategies of fairness preference and synergistic cooperation. Moreover, it will study in depth the principal–agent coordination and incentive strategies among Fresh Food Producers, Distributors, and CCL Service Providers. The goal is to incent CCL Service Providers to improve their freshness preservation efforts and achieve optimal overall benefits for the fresh food supply chain.

To sum up, the kernel of the mainstream research is inseparable from the freshness of fresh agricultural products, and research on the supply chain of fresh agricultural products from the perspective of information is mainly concentrated in the field of information sharing. However, comparisons on the degree of information completeness are lacking. In the research on the entrusted agent of the cold chain service, fewer scholars have conducted an in–depth analysis on the relationship of the entrusted agency, and even fewer related studies have been conducted on the pluralism of the participants' main bodies and the construction of their coordinating relationship. Combined with the above research basis, this study explores the principal–agent problem of cold chain service in the fresh agricultural products supply chain and takes producers, distributors, and cold chain service providers in the fresh agricultural products supply chain as research objects. It examines what kind of supply chain coordination strategy can be adopted to achieve the optimal incentive effect for cold chain service providers in order to maximize the freshness gain of producers and distributors in the fresh produce supply chain. In particular, this paper discusses the coordination strategies of upstream and downstream enterprises in CCL services under different information conditions from the perspective of the supply chain and considers the impact of upstream and downstream coordination strategies in the fresh produce supply chain on the level of freshness of the produce entrusted to the cold chain service provider. It likewise incorporates the theory of synergistic effect into the scope of the discussion of supply chain coordination strategies and builds a model under different levels of information by using the principal–agent framework. This paper broadens the application of synergy theory and principal–agent theory in the fresh produce supply chain as well as provides case studies and theoretical support for realizing the overall optimal benefits of the fresh produce supply chain.

## 3. Problem description and assumptions

### 3.1 Problem description

After fresh food producers harvest or collect their products, fresh food reach consumers through distribution. When consumers choose fresh food, freshness is often their top concern. However, Fresh Food Producers and Distributors often lack the technical means and transportation capabilities for product preservation. Therefore, they mostly consider entrusting third–party CCL Service Providers with ensuring freshness during transportation. As the principals in the CCL service, Fresh Food Producers and Distributors aim to obtain higher preservation

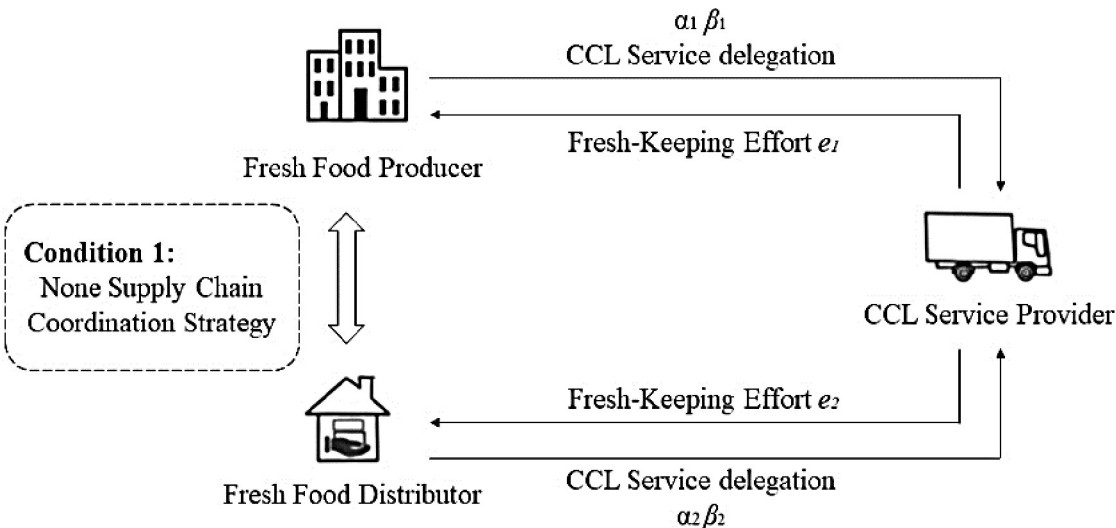

**Fig 1. Structure of the principal-agent base model of cold chain service of fresh produce supply chain.**

benefits at a lower cost. As the agent, the objective of CCL Service Providers is to achieve higher CCL service agency benefits at a lower preservation cost. Simultaneously, Fresh Food Producers and Distributors can choose coordination strategies to incentivize the CCL Service Providers, thereby maximizing the ultimate preservation benefits.

This study considers the CCL outsourcing and agency relationship between a Fresh Food Producer, Distributor, and CCL Service Provider. The Fresh Food Producer and Distributor in the fresh food supply chain entrust the fresh food preservation and transportation to the CCL Service Provider acting as an agent. By adjusting the supply chain coordination strategy, they aim to incentivize the CCL Service Provider to enhance their preservation efforts, ultimately maximizing the overall preservation benefits of the fresh food supply chain. Specifically, there are two scenarios. Scenario 1 considers the supply chain coordination strategy in establishing the basic model of the fresh food supply CCL service principal–agent (Fig 1). Scenario 2 considers the cooperative strategy in establishing the commissioning agent model of CCL service in the fresh food supply chain for the Fresh Food Producer and Distributor who jointly commission the CCL Service Provider (Fig 2).

The following assumptions are implemented in this research.

Assumption 1: The Fresh Food Producer and Distributor in the fresh food supply chain entrust their fresh food preservation and transportation business to the CCL Service Provider, thus establishing a tripartite subject in the CCL service principal–agent relationship. The Fresh Food Producer and Distributor in the fresh food supply chain are principals, and the CCL Service Provider is the agent.

Assumption 2: Based on the classic principal–agent theory of HM(Hygienic Motivator Theory) described in the references and the traditional assumption of rational individuals, it is assumed that the principal (i.e. the Fresh Food Producer and Distributor) is risk neutral and completely rational, and the agent (i.e. the CCL Service Provider) is completely rational and risk averse [49, 50].

Assumption 3: Assume, on the one hand that the freshness gains of the Fresh Food Producer and Distributor depend on the level of effort put in by the CCL Service Provider, and are also influenced, on the other hand, by uncertainties. Thus, the linear output function is denoted as $\pi_n^i = e\gamma + \theta$, where $\gamma > 0$ denotes the effort output coefficient; $\theta$ is an exogenous

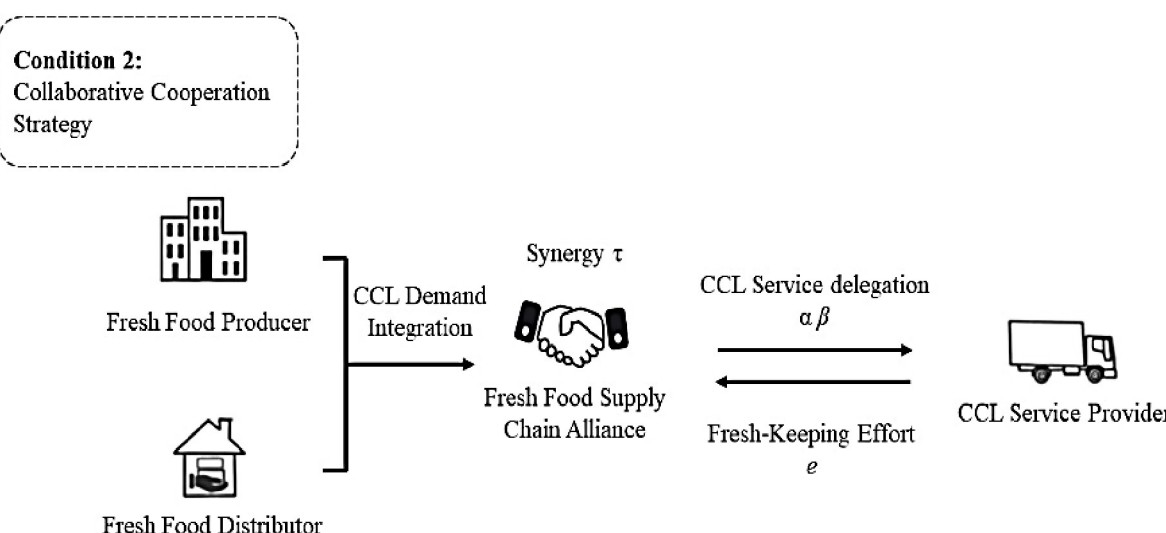

**Fig 2. Structure of the principal-agent model of cold chain service considering synergy effect.**

random variable, with $\theta \sim N(0,\sigma^2)$; and $e$ denotes the level of effort put in by the CCL Service Provider and the effort level of the freshness preservation task[51, 52].

Assumption 4: The Fresh Food Producer and Distributor must pay a certain cost to the CCL Service Provider. Given that the optimal incentive contract is in the linear form, $s(\pi) = \alpha + \beta\pi, \leq\beta\leq1$, it is assumed that the linear incentive payoffs $\alpha$ provided by the Fresh Food Producer and Distributor to the CCL Service Provider here is the fixed payoffs received by the CCL Service Provider, and $\beta$ is the incentive coefficient provided by the Fresh Food Producer and Distributor to the CCL Service Provider.

Assumption 5: The effort cost function of the CCL Service Provider is $C(e) = \frac{1}{2}be^2, b > 0$, where $b$ is the effort cost coefficient. This equation indicates that the marginal cost of effort increases gradually with the effort level, and the CCL Service Provider holds the retained utility $\omega,\omega>0$ for the Fresh Food Producer and Distributor, and the payment of the Fresh Food Producer and Distributor to the CCL Service Provider cannot be less than the retained utility [53].

Assumption 6: Costs in the supply chain are standardized to 0, except effect cost. This assumption is already quite common in existing literature, so we will not repeat it to prove it.

## 3.2 Parameters

The notations used in the mathematical model are shown in Table 1.

## 4. Mathematical formulation and solution

### 4.1 Cold chain logistics service principal–agent base model

The base model constructed based on the common principal–agent theory is shown in Fig 1. The Fresh Food Producer and Distributor independently decide on the intensity coefficient of their incentive towards the CCL Service Provider, while the latter strives to maximize their benefits and determine the level of effort to exert for the Fresh Food Producer and Distributor individually. Therefore, we obtain the expected net revenue functions for the Fresh Food Producer and Distributor as follow:

$$Y_n = E(\pi_n - s(\pi_n)) = -\alpha_n + (1 - \beta_n)e_n\gamma_n \tag{1}$$

**Table 1. Related parameter setting.**

| Variables | Parameters | Parameter descriptions |
|---|---|---|
| Exogenous variables Exogenous variables | $b$ | Cost of effort factor for CCL service providers |
| | $\gamma$ | Effort output factor of CCL service |
| | $\omega$ | Retained utility of CCL service |
| | $r$ | Absolute risk aversion factor for CCL service providers |
| | $\theta$ | Exogenous random variable |
| | $x$ | Actual money income |
| | $\alpha$ | Fixed compensation |
| | $\sigma^2$ | Variance of observable information |
| | $T$ | Coefficient of synergies |
| Decision variables | $e$ | Level of effort of CCL service providers |
| | $\beta$ | Incentive intensity factor for fresh food producers or distributors |
| Function | $\pi$ | Expected revenue function for fresh food producers or distributors |
| | $Y$ | Expected net revenue function for fresh food producers or Distributors |
| | $s$ | Expected return function of CCL service providers |
| | $Z$ | Expected net return function for CCL service providers |
| | $C$ | Effort cost function for CCL service providers |
| Top and bottom labels | $()_n^i$ | $i\in\{P,B\}$ represents the full information condition versus the incomplete information condition; $c$ represents the existence of synergy between the principals; $n\in\{1,2\}$, where 1 is a fresh food producer and 2 is a fresh food distributor. $P$ represents the base case for full information conditions, $B$ represents the base case for incomplete information conditions, $Pc$ represents the condition of collaborative cooperation among principals under conditions of full information, $Bc$ represents the condition of collaborative cooperation among principals under conditions of incomplete information. |

Then the expected net revenue functions of the Fresh Food Producer and Distributor are:

$$Y_1 = -\alpha_1 + (1 - \beta_1)e_1\gamma_1 \tag{2}$$

$$Y_2 = -\alpha_2 + (1 - \beta_2)e_2\gamma_2 \tag{3}$$

The net benefit of the CCL Service Provider is the incentive payoff it receives minus the cost of effort, and since they are risk averse, the utility function of the CCL Service Provider has an invariant risk aversion characteristic, $u(x) = -e^{-rx}$, where $r$ is the absolute risk aversion coefficient. $r>0$, and $x$ is the actual monetary income. The deterministic equivalent utility function of the CCL Service Provider is $E(x) - \frac{1}{2}r_n\beta_n^2\sigma_n^2$, where $E(x)$ is the expected utility function of the CCL Service Provider, and $\frac{1}{2}r_n\beta_n^2\sigma_n^2$ is the cost of risk.

Thus, the CCL Service Provider maximize the expected net benefit function equivalent to the deterministic equivalent utility function as

$$Z_n = E(s(\pi_n) - C(e_n)) = E[\alpha_n + \beta_n(e_n\gamma_n + \theta_n) - \frac{1}{2}b_ne_n^2 - \frac{1}{2}r_n\beta_n^2\sigma_n^2] \tag{4}$$

Then the expected net benefit functions of the CCL Service Provider from the Fresh Food Producer and Distributor, respectively are

$$Z_1 = \alpha_1 + \beta_1 e_1\gamma_1 - \frac{1}{2}b_1e_1^2 - \frac{1}{2}r_1\beta_1^2\sigma_1^2 \tag{5}$$

$$Z_2 = \alpha_2 + \beta_2 e_2 \gamma_2 - \frac{1}{2} b_2 e_2^2 - \frac{1}{2} r_2 \beta_2^2 \sigma_2^2 \tag{6}$$

The CCL Service Provider will choose to accept this contract only if its deterministic equivalent revenue is not less than $\omega_i$. Therefore, the participation constraint (*IR*) of the CCL Service Provider can be expressed as follows:

$$(IR)\alpha_1 + \beta_1 e_1 \gamma_1 - \frac{1}{2} b_1 e_1^2 - \frac{1}{2} r_1 \beta_1^2 \sigma_1^2 \geq \omega_1 \tag{7}$$

$$(IR)\alpha_2 + \beta_2 e_2 \gamma_2 - \frac{1}{2} b_2 e_2^2 - \frac{1}{2} r_2 \beta_2^2 \sigma_2^2 \geq \omega_2 \tag{8}$$

Under the incentive compatibility constraint (*IC*), the rational CCL Service Provider will choose the input level that maximizes the benefit. Hence, there is

$$(IC)e_1 \in \underset{e_1}{argmax} \left( \alpha_1 + \beta_1 e_1 \gamma_1 - \frac{1}{2} b_1 e_1^2 - \frac{1}{2} r_1 \beta_1^2 \sigma_1^2 \right) \tag{9}$$

$$(IC)e_2 \in \underset{e_2}{argmax} \left( \alpha_2 + \beta_2 e_2 \gamma_2 - \frac{1}{2} b_2 e_2^2 - \frac{1}{2} r_2 \beta_2^2 \sigma_2^2 \right) \tag{10}$$

Based on the analysis of the above assumptions, the following mathematical framework can be constructed using Eqs (7)–(12):

$$\underset{\alpha_1,\beta_1,e_1}{max} Y_1 = \max(-\alpha_1 + (1 - \beta_1)e_1\gamma_1) \tag{11}$$

$$\underset{\alpha_2,\beta_2,e_2}{max} Y_2 = \max(-\alpha_2 + (1 - \beta_2)e_2\gamma_2) \tag{12}$$

**4.1.1 Basic model solving under full information conditions.** When the Fresh Food Producer and Distributor can observe the effort level of the CCL Service Provider, the effort level of the latter is symmetric information. The only condition for the cooperation is that the CCL Service Provider can ensure that the income from the fresh food preservation task of the agent producer and distributor is not lower than the retained utility, and only the participation constraint (IR) of the CCL Service Provider is considered at this time.

Since the Fresh Food Producer and Distributor do not need to pay the CCL Service Provider more than the retained utility, the participation constraints (7) and (8) obtain the equals sign to obtain the fixed payoffs that the CCL Service Provider receives from the Fresh Food Producer and Distributor, respectively:

$$\alpha_1^P = \omega_1 - \beta_1^P e_1^P \gamma_1 + \frac{1}{2} b_1 (e_1^P)^2 + \frac{1}{2} r_1 (\beta_1^P)^2 \sigma_1^2 \tag{13}$$

$$\alpha_2^P = \omega_2 - \beta_2^P e_2^P \gamma_2 + \frac{1}{2} b_2 (e_2^P)^2 + \frac{1}{2} r_2 (\beta_2^P)^2 \sigma_2^2 \tag{14}$$

Substituting Eqs (13) and (14) into the expected net revenue functions of the Fresh Food

Producer and Distributor, respectively, yields

$$Y_1^P = -\omega_1 - \frac{1}{2}b_1(e_1^P)^2 - \frac{1}{2}r_1\left(\beta_1^P\right)^2\sigma_1^2 + e_1^P\gamma_1 \tag{15}$$

$$Y_2^P = -\omega_2 - \frac{1}{2}b_2(e_2^P)^2 - \frac{1}{2}r_2\left(\beta_2^P\right)^2\sigma_2^2 + e_2^P\gamma_2 \tag{16}$$

It is obtained by taking the partial derivative of $e_1^P$、$e_2^P$ in Eqs (15) and (16) and making it equal to 0. Then the optimal effort level of the CCL Service Provider under complete information is obtained as:

$$e_1^P* = \frac{\gamma_1}{b_1} \tag{17}$$

$$e_2^P* = \frac{\gamma_2}{b_2} \tag{18}$$

Such optimal effort level is obtained by taking the partial derivative of $\beta_1^P$、$\beta_2^P$ in Eqs (15) and (16) and making it equal to 0. Obtaining optimal incentive coefficients of the Fresh Food Producer and Distributor for the CCL Service Provider under full information conditions leads to: $\beta_1^P* = \beta_2^P* = 0$.

Substituting equations Eqs (17) and (18) into equations Eqs (13)–(16) yields the expected net returns of the Fresh Food Producer and Distributor under the full information condition and the fixed payoffs that the CCL Service Provider receives from the Fresh Food Producer and Distributor:

$$Y_1^P* = -\omega_1 + \frac{\gamma_1^2}{2b_1} \tag{19}$$

$$Y_2^P* = -\omega_2 + \frac{\gamma_2^2}{2b_2} \tag{20}$$

$$\alpha_1^P* = \omega_1 + \frac{\gamma_1^2}{2b_1} \tag{21}$$

$$\alpha_2^P* = \omega_2 + \frac{\gamma_1^2}{2b_1} \tag{22}$$

**4.1.2 Fundamental model solving under conditions of incomplete information.** In the actual cold chain service principal–agent relationship, as the principal, information asymmetry will lead to the inability to achieve Pareto optimality. At this time, when designing the incentive mechanism, it is necessary to consider the participation constraint (IR) and incentive compatibility constraint (IC) according to the CCL Service Provider.

Since the Fresh Food Producer and Distributor do not need to pay the CCL Service Provider more than the retained utility, the participation constraint Eqs (7) and (8) take the equal

sign to obtain the fixed payoff of the CCL Service Provider, respectively:

$$\alpha_1^B = \omega_1 - \beta_1^B e_1^B \gamma_1 + \frac{1}{2} b_1 (e_1^B)^2 + \frac{1}{2} r_1 (\beta_1^B)^2 \sigma_1^2 \tag{23}$$

$$\alpha_2^B = \omega_2 - \beta_2^B e_2^B \gamma_2 + \frac{1}{2} b_2 (e_2^B)^2 + \frac{1}{2} r_2 (\beta_2^B)^2 \sigma_2^2 \tag{24}$$

The optimal level of effort of the CCL Service Provider under incomplete information is obtained by taking the partial derivation of their effort $e_1$ and $e_2$ for the incentive compatibility constraints (9) and (10), making them equal to 0:

$$e_1^B * = \frac{\beta_1^B \gamma_1}{b_1} \tag{25}$$

$$e_2^B * = \frac{\beta_2^B \gamma_2}{b_2} \tag{26}$$

Substituting Eqs (23)–(24) into the expected net return functions of the Fresh Food Producer and Distributor, respectively, yields:

$$Y_1^B = -\omega_1 + \frac{(\beta_1^B)^2 \gamma_1^2}{2b_1} - \frac{r_1 (\beta_1^B)^2 \sigma_1^2}{2} + \frac{(1 - \beta_1^B) \beta_1^B \gamma_1^2}{b_1} \tag{27}$$

$$Y_2^B = -\omega_2 + \frac{(\beta_2^B)^2 \gamma_2^2}{2b_2} - \frac{r_2 (\beta_2^B)^2 \sigma_2^2}{2} + \frac{(1 - \beta_2^B) \beta_2^B \gamma_2^2}{b_2} \tag{28}$$

The partial derivatives and the second–order partial derivatives in Eqs (27) and (28) are obtained as follow:

$$\frac{\partial Y_1^B}{\partial \beta_1^B} = -r_1 \beta_1^B \sigma_1^2 + \frac{(1 - \beta_1^B) \gamma_1^2}{b_1} \quad \frac{\partial^2 Y_1^B}{\partial (\beta_1^B)^2} = -r_1 \sigma_1^2 - \frac{\gamma_1^2}{b_1} < 0 \tag{29}$$

$$\frac{\partial Y_2^B}{\partial \beta_2^B} = -r_2 \beta_2^B \sigma_2^2 + \frac{(1 - \beta_2^B) \gamma_2^2}{b_2} \quad \frac{\partial^2 Y_2^B}{\partial (\beta_2^B)^2} = -r_2 \sigma_2^2 - \frac{\gamma_2^2}{b_2} < 0 \tag{30}$$

We find that the second–order partial derivative exists and is less than 0, so there is a maximum value of the objective function under the condition of incomplete information. In addition, the optimal incentive coefficients of the Fresh Food Producer and Distributor under the incomplete information model can be found:

$$\beta_1^B * = \frac{\gamma_1^2}{r_1 \sigma_1^2 b_1 + \gamma_1^2} \tag{31}$$

$$\beta_2^B * = \frac{\gamma_2^2}{r_2 \sigma_2^2 b_2 + \gamma_2^2} \tag{32}$$

Substituting Eqs (31) and (32) into $e_1^B *$ and $e_2^B *$, respectively, yields the simplified optimal level

of effort of the CCL Service Provider in the incomplete information model:

$$e_1^B* = \frac{\gamma_1{}^3}{(r_1\sigma_1^2 b_1 + \gamma_1{}^2)b_1} \tag{33}$$

$$e_2^B* = \frac{\gamma_2{}^3}{(r_2\sigma_2^2 b_2 + \gamma_2{}^2)b_2} \tag{34}$$

We can yield the expected net returns of the Fresh Food Producer and Distributors under the incomplete information condition and the fixed payoffs that the CCL Service Provider receives from the Fresh Food Producer and Distributor as:

$$Y_1^B* = \frac{-2b_1^2 r_1 \omega_1 \sigma_1^2 - 2b_1 \gamma_1{}^2 \omega_1 + \gamma_1{}^4}{2(r_1\sigma_1^2 b_1 + \gamma_1{}^2)b_1} \tag{35}$$

$$Y_2^B* = \frac{-2b_2^2 r_2 \omega_2 \sigma_2^2 - 2b_2 \gamma_2{}^2 \omega_2 + \gamma_2{}^4}{2(r_2\sigma_2^2 b_2 + \gamma_2{}^2)b_2} \tag{36}$$

$$\alpha_1^B* = \omega_1 - \frac{\gamma_1{}^6}{2(r_1\sigma_1^2 b_1 + \gamma_1{}^2)^2 b_1} + \frac{r_1\gamma_1{}^4\sigma_1^2}{2(r_1\sigma_1^2 b_1 + \gamma_1{}^2)^2} \tag{37}$$

$$\alpha_2^B* = \omega_2 - \frac{\gamma_2{}^6}{2(r_2\sigma_2^2 b_2 + \gamma_2{}^2)^2 b_2} + \frac{r_2\gamma_2{}^4\sigma_2^2}{2(r_2\sigma_2^2 b_2 + \gamma_2{}^2)^2} \tag{38}$$

Proposition 1: Given that $e_n^P* - e_n^B* > 0$, the freshness effort level decision of the CCL Service Provider satisfies $e_n^P* > e_n^B*$.

Proposition 1 shows that in the base model and the cooperative model, the effort levels of the CCL Service Provider under the complete information condition are greater than those under the incomplete information condition. This is because under the complete information condition, because the Fresh Food Producer and Distributor fully observe the information on the freshness effort level of the CCL Service Provider, there is no room for the latter to take advantage of information laziness. Thus, the CCL Service Provider will exert more effort than under the incomplete information condition.

Proposition 2: According to Eqs (17)–(18) and (25)–(26), it can be concluded that $\frac{\partial e^*}{\partial \gamma} > 0, \frac{\partial e^*}{\partial b} < 0$. Therefore, the optimal effort level of the CCL Service Provider is positively correlated with the effort output coefficient and negatively correlated with the effort cost coefficient.

Proposition 2 shows that the effort level of the CCL Service Provider is determined by the interaction between effort output and effort cost and that the CCL Service Providers can enhance their optimal effort level by choosing tasks with lower effort cost and higher effort output. Therefore, for the Fresh Food Producer and Distributor, the difficulty and output of the preservation tasks entrusted by them are the key to determining the level of preservation efforts paid by the CCL Service Provider.

Proposition 3: According to $\beta_1^P* = \beta_2^P* = \beta^{Pc^*} = 0$, the incentive coefficient is 0 under the condition of complete information.

Proposition 3 shows that under the condition of complete information, the supply chain incentive strategy fails. Since the Fresh Food Producer and Distributor can observe the freshness preservation efforts of the CCL Service Provider, there is no room for the latter to take

advantage of information to encroach on the interests of the Fresh Food Producer and Distributor.

Proposition 4: According to a simplification Eqs (31)–(32), it can be obtained that $\frac{\partial \beta^*}{\partial \gamma} > 0, \frac{\partial \beta^*}{\partial b} < 0, \frac{\partial \beta^*}{\partial r} < 0$. Hence, under conditions of incomplete information, the incentive coefficients of the Fresh Food Producer and Distributor are positively related to the coefficients of effort output of the CCL Service Provider and negatively related to the coefficients of effort cost and risk aversion.

Proposition 4 shows that the incentive coefficients set by the Fresh Food Producer and Distributor are determined by the interaction of the output of effort, the cost of effort, and the degree of risk aversion.

Proposition 5: According to the simplification of Eqs (19)–(20) and (35)–(36), it can be concluded that $\frac{\partial Y}{\partial \gamma} > 0, \frac{\partial Y}{\partial b} < 0$. Hence, the expected net benefits of the Fresh Food Producer and Distributor are negatively related to the coefficient of effort cost of the CCL Service Provider and positively related to their effort output coefficients.

Proposition 5 shows that under any information condition or supply chain incentive strategy, the expected net benefits of the Fresh Food Producer and Distributor are determined by the interaction between the effort output and effort cost of the CCL Service Provider. This finding suggests that in the cold chain service principal–agent relationship in the fresh food supply chain, the principal should also adopt some strategies to help the CCL Service Provider reduce the cost of cold chain transportation and improve its output benefit. The client can also get higher net income from it, which ultimately realizes the win–win situation for both parties in the cold chain service principal–agent dynamic in the fresh food supply chain.

Proposition 6: Given that $Y_n^B * - Y_n^P * > 0$, the expected net returns of the Fresh Food Producer and Distributor satisfy $Y_n^B * > Y_n^P *$.

Proposition 6 shows that whether it is between the Fresh Food Producer and Distributor or between them and cold chains, the degree of transparency and openness of information will ultimately affect each of their final returns. Strengthening information exchange is the key to effectively improving the freshness of fresh food.

## 4.2 Collaborative cold chain logistics service delegation model

According to the synergy theory proposed by Haken [54], collaboration among individuals creates collective benefits or overall effects. It is assumed that the formation of an alliance between the Fresh Food Producer and Distributor will generate additional collective benefits, denoted as the synergy benefit coefficient $\tau$. A larger value $\tau$ indicates closer collaboration between the Fresh Food Producer and Distributor, resulting in greater economies of scale and synergy benefits. Specifically, when $\tau = 0$, the synergy benefit coefficient between the Fresh Food Producer and Distributor is 0, indicating the absence of a collaborative relationship. Therefore, the relationship between the fresh food supply chain alliance and the CCL Service Provider can be seen as a single delegation agency relationship. The linear profit level of the fresh food supply chain alliance is denoted as $P = e(\gamma+\tau)+\theta$. The cold chain service delegation model based on common agency theory and synergy theory is illustrated in Fig 2. The remaining assumptions are the same as in the base model.

Consider that the linear profit level of the fresh produce supply chain alliance is $P = e(\gamma+\tau)+\theta$. Likewise, we can conclude that:

The expected net profit function of the fresh food supply chain alliance under collaborative cooperation is:

$$Y^c = E(\pi - s(\pi)) = -\alpha^c + e^c(1 - \beta^c)(\gamma + \tau) \tag{39}$$

Under collaborative cooperation, the expected net profit function of the CCL Service Provider can be expressed as:

$$Z^c = E(s(\pi) - C(e^c)) = \alpha^c + e^c(\gamma + \tau)\beta^c - \frac{1}{2}b(e^c)^2 - \frac{1}{2}r(\beta^c)^2\sigma^2 \tag{40}$$

Under collaborative cooperation, the participation constraint (IR) for the CCL Service Provider can be represented as:

$$Z^c = \alpha^c + e^c(\gamma + \tau)\beta^c - \frac{1}{2}b(e^c)^2 - \frac{1}{2}r(\beta^c)^2\sigma^2 \geq \omega \tag{41}$$

Under collaborative cooperation, the incentive compatibility constraint (IC) for the CCL Service Provider can be represented as:

$$e^c \in \underset{e^c}{argmax}(\alpha^c + e^c(\gamma + \tau)\beta^c - \frac{1}{2}b(e^c)^2 - \frac{1}{2}r(\beta^c)^2\sigma^2) \tag{42}$$

According to the analysis based on the above assumptions, we can construct the following mathematical framework, which is represented by Eqs (43)–(45):

$$\underset{\alpha,\beta,e}{max}\, Y^c = \max(-\alpha^c + e^c(1 - \beta^c)(\gamma + \tau)) \tag{43}$$

$$(IR)Z^c = \alpha^c + e^c(\gamma + \tau)\beta^c - \frac{1}{2}b(e^c)^2 - \frac{1}{2}r(\beta^c)^2\sigma^2 \geq \omega \tag{44}$$

$$(IC)e^c \in \underset{e^c}{argmax}(\alpha^c + e^c(\gamma + \tau)\beta^c - \frac{1}{2}b(e^c)^2 - \frac{1}{2}r(\beta^c)^2\sigma^2) \tag{45}$$

**4.2.1 Model solution under full information condition with consideration of collaborative cooperation.** The effort level of the CCL Service Provider can be fully observed under the full information condition. Therefore, the CCL Service Provider can only ensure that the income obtained from collaborating with the fresh food supply chain alliance is not lower than its retained utility. In this case, only the participation constraint (IR) of the CCL Service Provider is considered.

Similarly, when the participation constraint (IR) is met with equality, we can obtain the fixed remuneration for the CCL Service Provider as follow:

$$\alpha^{Pc} = \omega - e^{Pc}(\gamma + \tau)\beta^{Pc} + \frac{1}{2}b(e^{Pc})^2 + \frac{1}{2}r(\beta^{Pc})^2\sigma^2 \tag{46}$$

By substituting Eq (46) into the expected net profit function of the fresh food supply chain alliance, we obtain:

$$Y^{Pc} = -\omega + e^{Pc}(\gamma + \tau) - \frac{1}{2}b(e^{Pc})^2 - \frac{1}{2}r(\beta^{Pc})^2\sigma^2 \tag{47}$$

Taking the partial derivative of Eq (47) and setting it equal to 0, we obtain the optimal effort level for the CCL Service Provider under complete information and collaborative cooperation:

$$e^{Pc*} = \frac{\gamma + \tau}{b} \tag{48}$$

Taking the partial derivative of Eq (47) and setting it equal to 0, we can obtain the optimal

incentive coefficient for the enterprise under the condition of complete information and collaborative cooperation $\beta^{Pc*} = 0$.

Above all, we can obtain the expected net profit of the fresh food supply chain alliance and the fixed remuneration for the CCL Service Provider under the condition of complete information:

$$Y^{Pc*} = -\omega + \frac{(\gamma + \tau)^2}{2b} \tag{49}$$

$$\alpha^{Pc*} = \omega + \frac{(\gamma + \tau)^2}{2b} \tag{50}$$

**4.2.2 Model solution considering collaborative cooperation under incomplete information.** Under the condition of incomplete information, it is necessary to consider maximizing both the expected net profit of the fresh food supply chain alliance and maximizing the expected net profit of the CCL Service Provider.

We can obtain the fixed remuneration for the CCL Service Provider as follow:

$$\alpha^{Bc} = \omega - e^{Bc}(\gamma + \tau)\beta^{Bc} + \frac{1}{2}b(e^{Bc})^2 + \frac{1}{2}r(\beta^{Bc})^2\sigma^2 \tag{51}$$

In the same way, we can obtain the optimal effort level of the CCL Service Provider in the case of collaborative cooperation under incomplete information:

$$e^{Bc*} = \frac{(\gamma + \tau)\beta^{Bc}}{b} \tag{52}$$

By substituting Eqs (51)–(52) into the expected net profit function of the fresh food supply chain alliance, we can obtain the following result:

$$Y^{Bc} = -\omega + \frac{(\gamma + \tau)^2(\beta^{Bc})^2}{2b} - \frac{r(\beta^{Bc})^2\sigma^2}{2} + \frac{(\gamma + \tau)^2(1 - \beta^{Bc})\beta^{Bc}}{b} \tag{53}$$

To derive the partial derivative and second–order partial derivative of Eq (53) concerning $\beta$, we can calculate the following:

$$\frac{\partial Y^{Bc}}{\partial \beta^{Bc}} = -r(\beta^{Bc})^2\sigma^2 + \frac{(\gamma + \tau)^2(1 - \beta^{Bc})}{b} \tag{54}$$

$$\frac{\partial^2 Y^{Bc}}{\partial \beta^2} = -r\sigma^2 - \frac{(\gamma + \tau)^2}{b} \tag{55}$$

Based on the information provided, the second–order partial derivative exists and is negative. This finding implies that in the case of collaborative cooperation under incomplete information, the objective function has a maximum. Thus, we can solve for the optimal incentive coefficient for the enterprises in this scenario:

$$\beta^{Bc*} = \frac{(\gamma + \tau)^2}{r\sigma^2 b + (\gamma + \tau)^2} \tag{56}$$

By substituting Eq (57) into Eq (52), we can obtain the simplified expression for the optimal effort level of the CCL Service Provider in the case of collaborative cooperation under

incomplete information:

$$Y^{Bc^*} = \frac{(\gamma + \tau)^3}{(r\sigma^2 b + (\gamma + \tau)^2)b} \tag{57}$$

Above all, we can obtain the maximum expected net profit of the fresh food supply chain alliance and the fixed reward obtained by the CCL Service Provider under the condition of incomplete information:

$$Y^{Bc*} = -\omega + \frac{(\gamma + \tau)^6}{2(r\sigma^2 b + (\gamma + \tau)^2)b} + \frac{r(\gamma + \tau)^4 \sigma^2}{2(r\sigma^2 b + (\gamma + \tau)^2)} \tag{58}$$

$$\alpha^{Bc^*} = \omega - \frac{(\gamma + \tau)^6}{2(r\sigma^2 b + (\gamma + \tau)^2)b} + \frac{r(\gamma + \tau)^4 \sigma^2}{2(r\sigma^2 b + (\gamma + \tau)^2)} \tag{59}$$

Proposition 7: When $\gamma = \gamma_1 = \gamma_2, b = b_1 = b_2$, it can be concluded that $e^{Pc^*} - e_1^P* = \frac{\gamma}{b}$ and $e^{Pc^*} - e_2^P* = \frac{\gamma}{b}$. Therefore, the effort level of the CCL Service Provider satisfies $e^{Pc^*} > e_1^P*$, $e^{Pc^*} > e_2^P*$.

Proposition 7 suggests that under perfect information conditions when the effort–output ratio and effort–cost ratio of the CCL Service Providers remains at the same level, the effort level of the CCL Service Provider is higher when a fresh food supply chain alliance is established between the Fresh Food Producer and Distributors compared to when no alliance relationship is formed. This implies that building a fresh food supply chain alliance is beneficial for enhancing the preservation effort level of the CCL Service Provider.

Proposition 8: When $\gamma = \gamma_1 = \gamma_2$, $b = b_1 = b_2, r = r_1 = r_2$, and $\sigma = \sigma_1 = \sigma_2$, it can be concluded that $\beta^{Bc^*} - \beta_1^B* = \beta^{Bc^*} - \beta_2^B* > 0$. Thus, the incentive decisions of the Fresh Food Producer and Distributor satisfy: $\beta^{Bc^*} > \beta_1^B*$ and $\beta^{Bc^*} > \beta_2^B*$.

Proposition 8 indicates that under conditions of incomplete information, when the effort–output ratio, effort–cost ratio, variance, and risk aversion coefficient of the CCL Service Provider remain at the same levels, the incentive coefficient for cooperation between the Fresh Food Producers and Distributor is higher than when there is no cooperation. This means that building a fresh food supply chain alliance enhances the incentive toward the CCL Service Provider. Establishing an alliance between the Fresh Food Producer and Distributor creates economies of scale or collective effects that can better motivate the CCL Service Provider to improve its CCL and transportation capabilities.

Proposition 9: Simplification yields $\frac{\partial e^{Pc^*}}{\partial \tau} > 0, \frac{\partial e^{Bc^*}}{\partial \tau} > 0, \frac{\partial e^*}{\partial \tau} > 0$. A positive correlation exists between the optimal effort level of the cold chain service providers and the synergy coefficient in the supply chain coordination strategy under collaborative cooperation between the Fresh Food Producer and Distributor.

Proposition 9 suggests that the higher the level of collaboration and closeness of the relationship between the Fresh Food Producer and Distributor, the higher the synergy coefficient, which encourages the CCL Service Provider to exert a higher preservation effort level.

Proposition 10: Similar to Proposition 2, simplification yields $\frac{\partial \beta^{Bc^*}}{\partial \tau} > 0$. Under conditions of perfect information, there is no relationship between the incentive and synergy coefficients. Under conditions of incomplete information, there is a positive correlation between the incentive and synergy coefficients.

Proposition 10 states that under conditions of perfect information, the incentive mechanism is ineffective, and thus, collaborative cooperation within the fresh food supply chain

alliance does not affect the incentive mechanism. However, under conditions of incomplete information, a higher synergy coefficient between the Fresh Food Producer and Distributor can incentivize the CCL Service Provider to improve their service level, resulting in a higher intensity of incentives.

Proposition 11: $\frac{\partial Y^{Pc*}}{\partial \tau} > 0$ and $\frac{\partial Y^{Bc*}}{\partial \tau} > 0$, it can be concluded that $\frac{\partial Y}{\partial \tau} > 0$. In the supply chain coordination strategy under collaborative cooperation between the Fresh Food Producer and Distributor, the expected net income of the Fresh Food Producer and Distributor is positively correlated with the synergy coefficient.

Proposition 11 indicates that after establishing a collaborative relationship between the Fresh Food Producer and Distributor, the level of synergy between them has an impact on their expected net income. The higher the level of collaboration and closeness of cooperation between Fresh Food Producer and Distributor, the higher the expected net income.

Proposition 12: When $\gamma = \gamma_1 = \gamma_2, b = b_1 = b_2$, and $Y^{Pc*} > Y_1^P* + Y_2^P*$, it can be concluded that $\frac{(\gamma+\tau)^2}{2b} > \frac{\gamma^2}{2b_1} > \frac{\gamma^2}{2b_2}, \tau > (\sqrt{2} - 1)\gamma$, Thus, if $\tau > (\sqrt{2} - 1)\gamma$, then the expected income of the Fresh Food Producer and Distributor satisfies: $Y^{Pc*} > Y_1^P* + Y_2^P*$. Likewise, if $0 < \tau < (\sqrt{2} - 1)\gamma$, the expected income of the Fresh Food Producer and Distributor satisfies: $Y^{Pc*} < Y_1^P* + Y_2^P*$.

Proposition 12 indicates that the Fresh Food Producer and Distributor do not form a fresh food supply chain alliance with each other to immediately obtain greater freshness benefits,

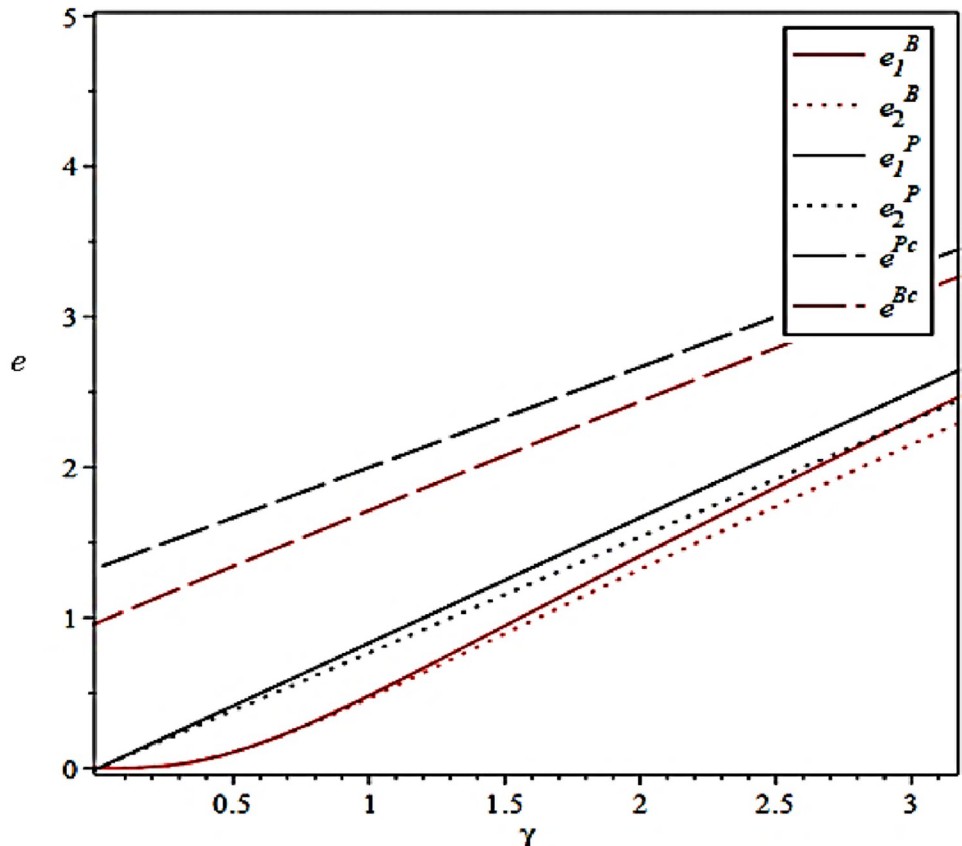

**Fig 3. Effect of effort output coefficient on level of effort.**

and the level of benefits is jointly determined by the interactions between effort output, effort cost, and synergy coefficient. Therefore, when the Fresh Food Producer and Distributor consider whether to establish a synergistic alliance, they need to consider not only the actual effort output and effort cost of the cold chain service providers, but also the actual level of synergistic cooperation that can be achieved after the synergistic cooperation.

## 5. Case study

This paper takes the cold chain transportation of the fruit route of Company S as a case study. High cold–chain transportation costs are incurred during the refrigeration and transportation of lychee routes. Buyers are more dominant because such relatively expensive seasonal fruits are not necessities. In this study, the parameters of the Fresh Food Producer and Distributor–dominated fresh food cold–chain transportation agency model are set in the numerical analysis.

According to the realities of the general environment and of the background of S Company S's agent of lychee cold chain transportation, the corresponding parameters take the value of the situation as follows: $b_1 = 1.2, b_2 = 1.3, b = 1.5, \gamma_1 = 1.3, \gamma_2 = 1.6,$

$\gamma = 1.4, \omega_1 = 1, \omega_2 = 1, \omega = 1, r_1 = 0.6, r_2 = 0.5, r = 0.7, \sigma_1^2 = 1, \sigma_2^2 = 1, \sigma^2 = 1$, and $\tau = 2$.

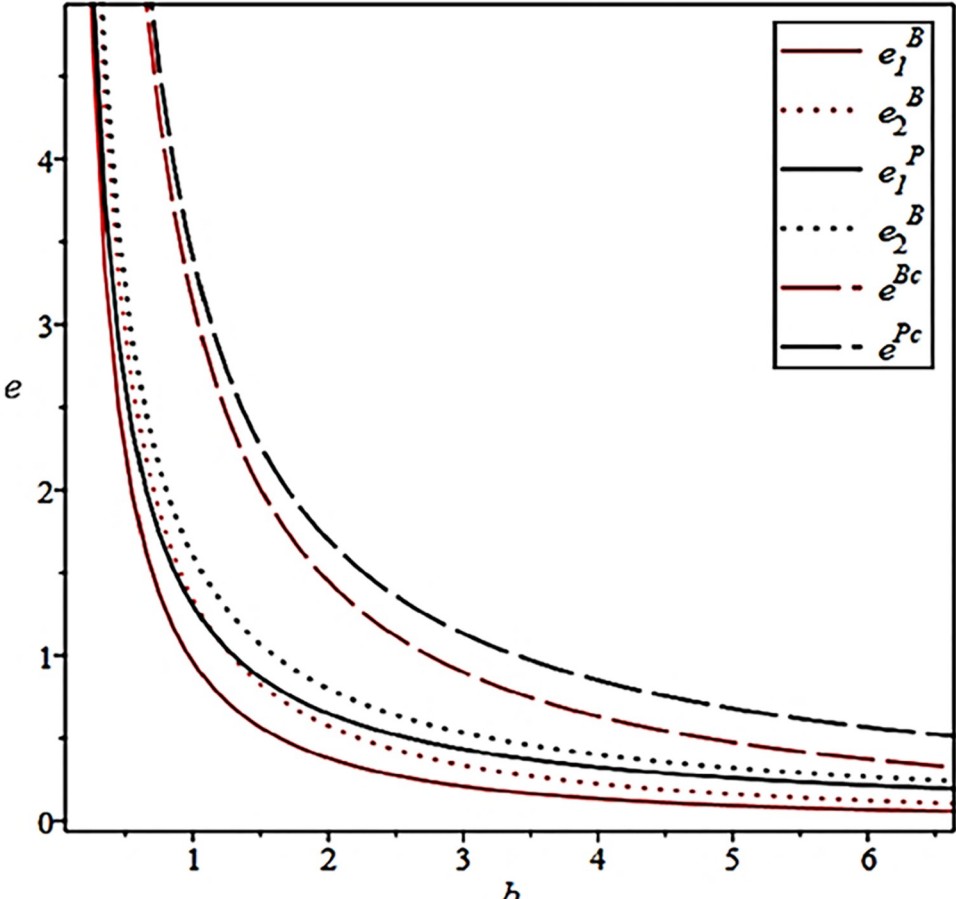

**Fig 4. Effect of effort cost coefficients on effort levels.**

The effect of the effort output coefficients of the CCL Service Provider on their effort levels under different scenarios is shown in Fig 3.

Fig 3 verifies Propositions 1 and 2. The CCL Service Provider will select the high–value business according to the contract value and the credibility of the enterprise proposed by the principal before taking over the preservation business. After the CCL Service Provider undertake the preservation business, they will also be more inclined to prioritize the business with higher output benefits, thus enabling it to obtain greater economic benefits. When effort output is kept within a certain range, the collaborative efforts between the Fresh Food Producer and Distributor can lead the CCL Service Provider to deliver high–quality preservation services. This outcome is likely because the cooperation between the Fresh Food Producer and Distributor expands the market scale and enhances reputation.

The influence of the effort cost coefficient of the CCL Service Provider on their effort levels under different scenarios is shown in Fig 4.

Fig 4 also verifies Propositions 1 and 2. The CCL Service Provider care about the input–to–output ratio of their agency business. Therefore, increasing economies of scale, optimizing the cold chain transportation network, and reducing redundant costs are effective strategies for the CCL Service Provider to increase efficiency and reduce costs.

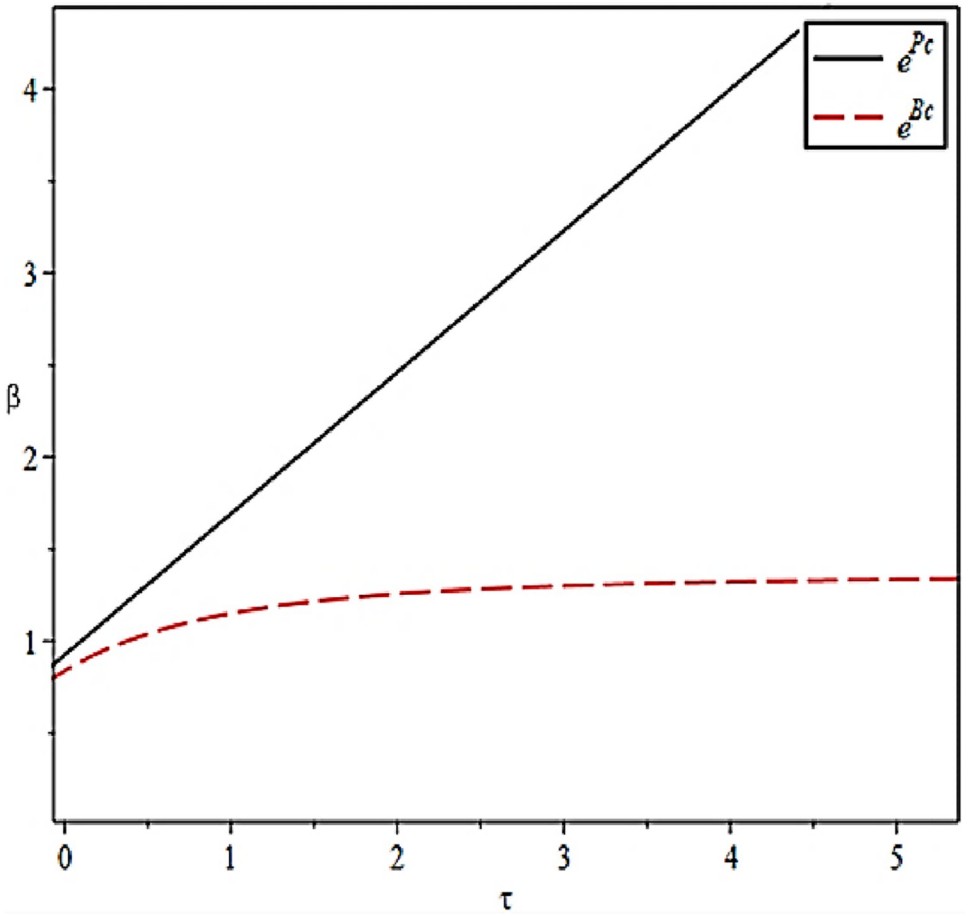

**Fig 5. Impact of synergy coefficients on the level of effort.**

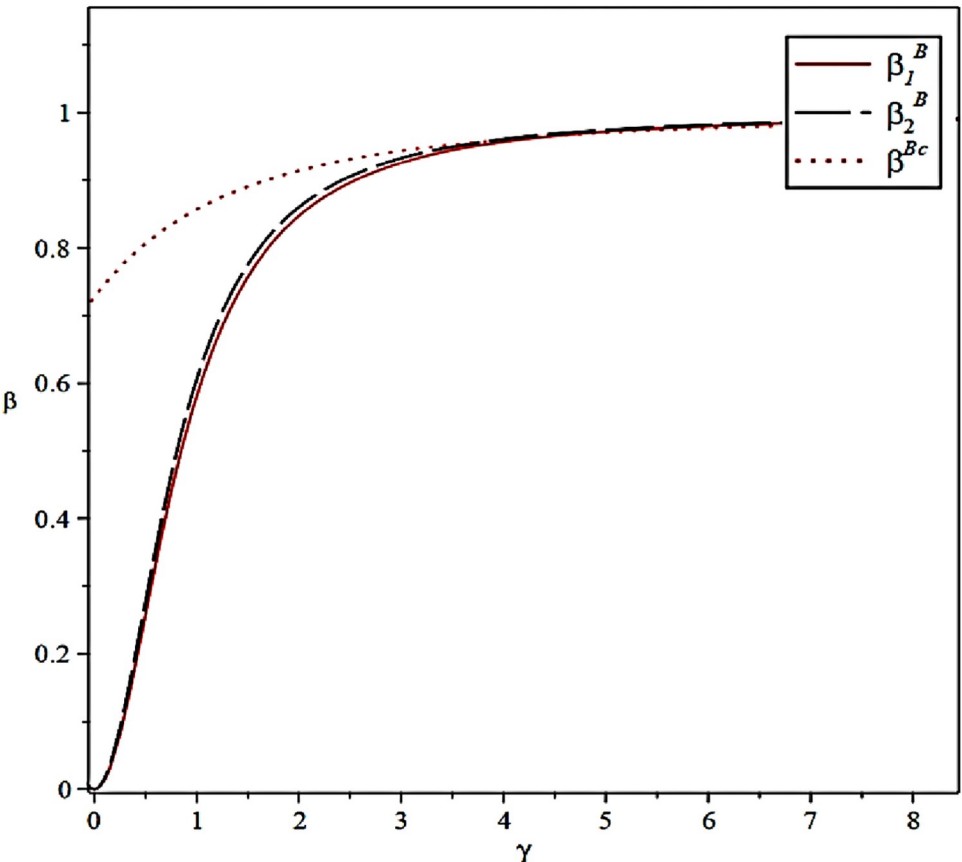

**Fig 6. Effect of effort-output coefficients on incentive strategies.**

The impact of the coefficient of synergistic benefits between the Fresh Food Producer and Distributor on the level of effort of the CCL Service Provider under different scenarios is shown in Fig 5.

Fig 5 corresponds to Proposition 9. In the case of equal coefficients of synergistic benefits of fresh food supply chain alliance, the fresh food supply chain alliance, under the condition of complete information, can obtain a higher level of freshness preservation from the CCL Service Provider. At the same time, the coefficient of synergistic benefit under the condition of complete information is found to be linearly correlated with the level of effort, while under the condition of incomplete information, the synergistic benefit and effort level do not show a single linear correlation.

The effect of effort output coefficients of the CCL Service Provider on the incentive level of the Fresh Food Producer and Distributor under different scenarios is shown in Fig 6.

Fig 6 corresponds to Proposition 4. This result suggests that in the cold chain principal–agent relationship, the economic costs that both parties are willing to pay are determined by each other, and in the case that the CCL Service Provider is willing to pay a higher effort output, the cold chain agent will also be more inclined to pay higher incentive costs.

The effect of the effort cost coefficient of the CCL Service Provider on the incentive level of the Fresh Food Producer and Distributor under different scenarios is shown in Fig 7.

Fig 7 can be related to Proposition 4. When effort output is kept within a certain range, the collaborative efforts between the Fresh Food Producers and Distributor can lead the cold

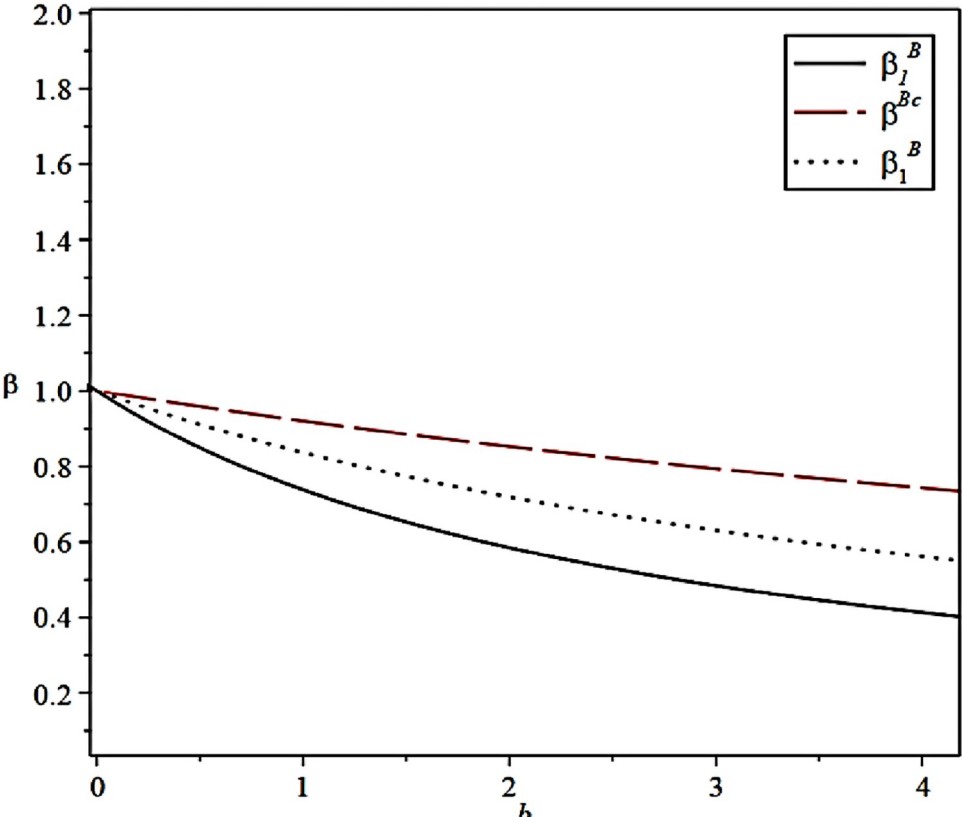

**Fig 7. Effect of cost of effort coefficient on incentive strategies.**

chain service provider to deliver high–quality preservation services. This outcome is likely because the cooperation between the Fresh Food Producer and Distributor expands the market scale and enhances the reputation level.

The effects of risk aversion coefficients of the CCL Service Provider on the incentive levels of the Fresh Food Producer and Distributor in different scenarios are shown in Fig 8.

This finding corresponds to Proposition 4. When the risk aversion coefficient of the CCL Service Provider gradually increases, the incentive intensity of the Fresh Food Producer and Distributor to the CCL Service Provider will gradually decrease, which indicates that the CCL Service Provider is more inclined to avoid risks. Meanwhile, the Fresh Food Distributor under incomplete information conditions always produces better incentive effects under the same risk aversion coefficient.

The impact of the coefficient of synergies between the Fresh Food Producer and Distributor on their incentive levels in different scenarios is shown in Fig 9.

Fig 9 verifies Proposition 10. It indicates that under the condition of incomplete information, the synergistic benefits and incentive strategies do not show a single linear correlation; In particular, the synergistic benefits of the fresh food supply chain alliance have a more significant effect on the incentive of the CCL Service Provider in the initial stage. Furthermore, when accompanied by the closer cooperation between the Fresh Food Producer and Distributor, which forms an inextricably integrated whole, the incentive effect of such cooperation for CCL Service Providers will become increasingly stable.

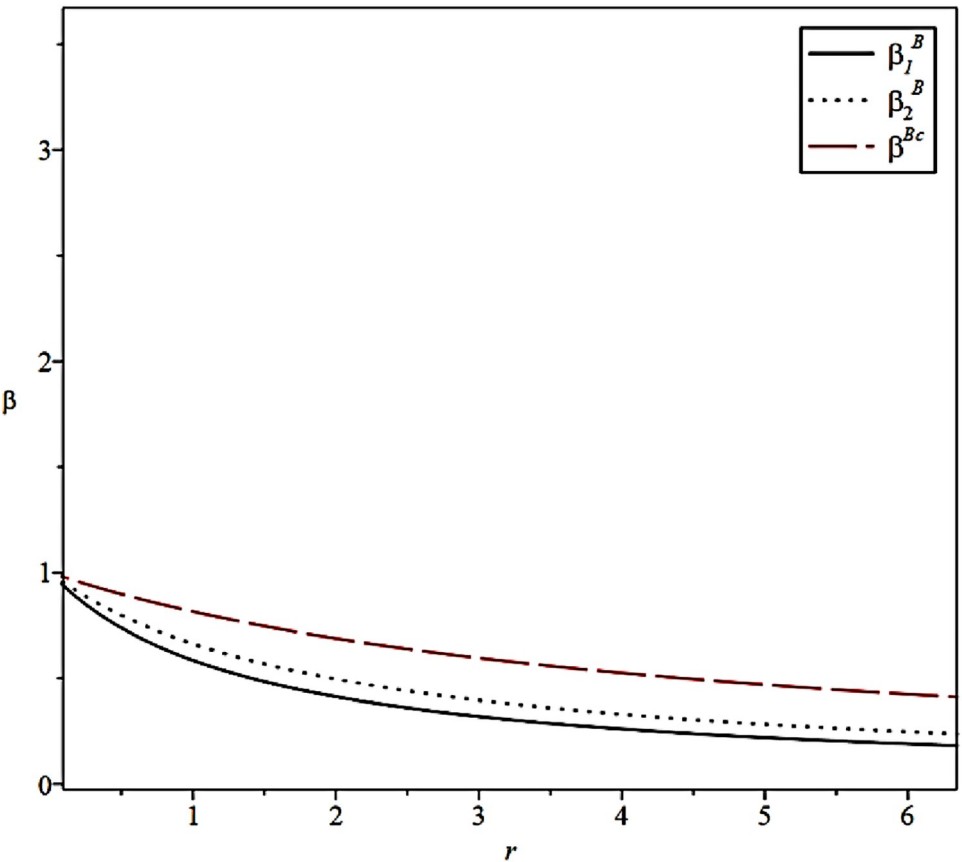

**Fig 8. Effect of risk aversion coefficient on incentive strategies.**

The impact of synergy coefficients between the Fresh Food Producers and Distributor on their incentive levels under different scenarios is shown in Fig 10.

Fig 10 is similar to the conclusion of Proposition 11. In the early stage of forming a fresh food supply chain alliance, both upstream and downstream participants will pay certain basic costs. However, in the long run, as the upstream and downstream participants of the fresh food supply chain cooperate more closely, the degree of synergy is improved, which can bring higher preservation benefits for the fresh food supply chain alliance.

## 6. Conclusion and remarks

### 6.1 Conclusion

Fresh food is perishable and prone to deterioration, and thus has high requirements for the timeliness of transportation and the freshness preservation environment. Accordingly, fresh food and sales enterprises usually outsource their non–core business to a professional CCL Service Provider. To incentivize CCL Service Providers to enhance their preservation efforts and ultimately maximize the overall preservation benefits of the fresh food supply chain, this study investigates a CCL service principal–agent problem in the fresh food supply chain involving tripartite participation of Fresh Food Producers, Distributors, and CCL Service Providers, Thus, the CCL service principal–agent model under different scenarios is established. The optimal level of preservation effort, optimal incentive coefficient, and highest preservation gain under different situations are obtained through the calculation of examples.

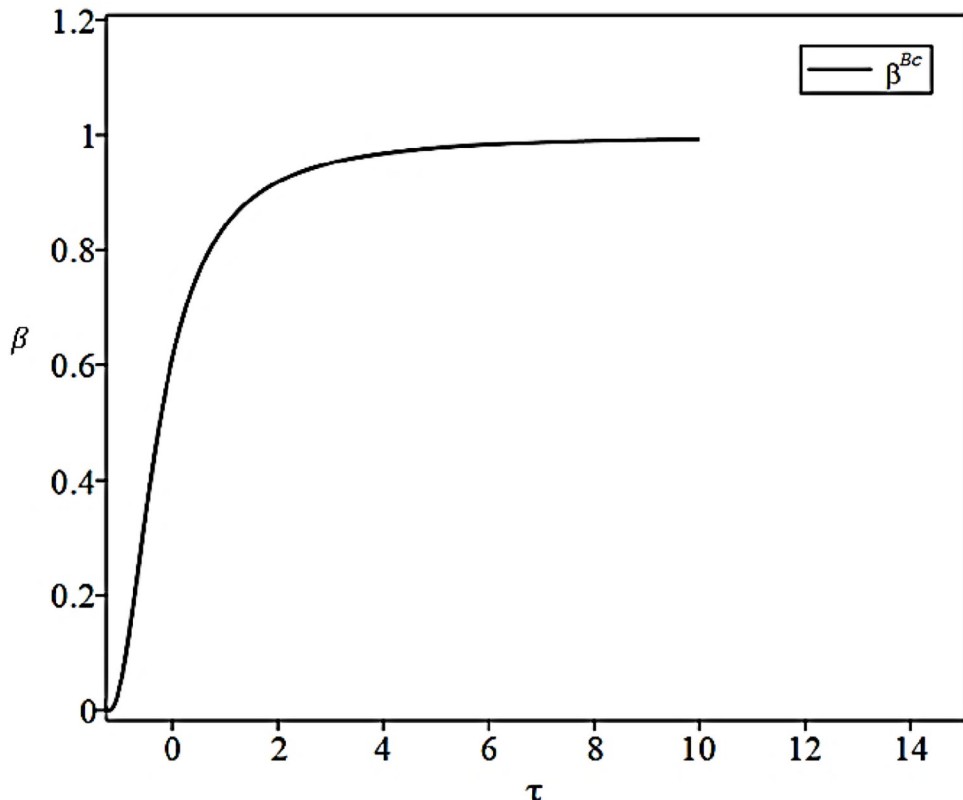

**Fig 9. Effect of synergy benefit coefficients on incentive strategies.**

To sum up, we can draw the following conclusions:

(1) In the basic and cooperative model, the effort level of the CCL Service Provider and the freshness preservation benefits of the Fresh Food Producer and Distributor are higher than those in the incomplete information condition. This result shows that the transparency and openness of information have a greater impact on the freshness preservation benefits of the fresh food CCL service consignors. (2) Under the condition of complete information, the freshness preservation incentive strategy of the Fresh Food Producer and Distributor fails, and the incentive coefficients are not related to the effort output and effort cost coefficients. Under the condition of incomplete information, the incentive coefficient is positively related to the coefficient of effort output and negatively related to the coefficient of effort cost and the coefficient of risk aversion. We find that the incentive strategy is affected by the interaction of effort output, effort cost, and risk aversion degree. Therefore, in practice, the fresh food incentive strategy needs to be flexibly adjusted according to the performance of the CCL Service Provider. (3) The freshness gains of the Fresh Food Producer and Distributor are negatively correlated with the effort level and effort cost of the CCL Service Provider and positively correlated with the effort output. This means that when the latter accepts tasks with lower effort costs and higher effort outputs, they can increase their effort levels and enhance the freshness preservation benefits of the Fresh Food Producers and Distributor simultaneously. (4) When the Fresh Food Producers and Distributor collaborate, the level of preservation efforts of the CCL Service Provider and the incentive coefficients of the former are higher than those under the base model under the same information conditions. Moreover, when the Fresh Food Producer and Distributor cooperate more closely and the degree of synergy is higher, the level of

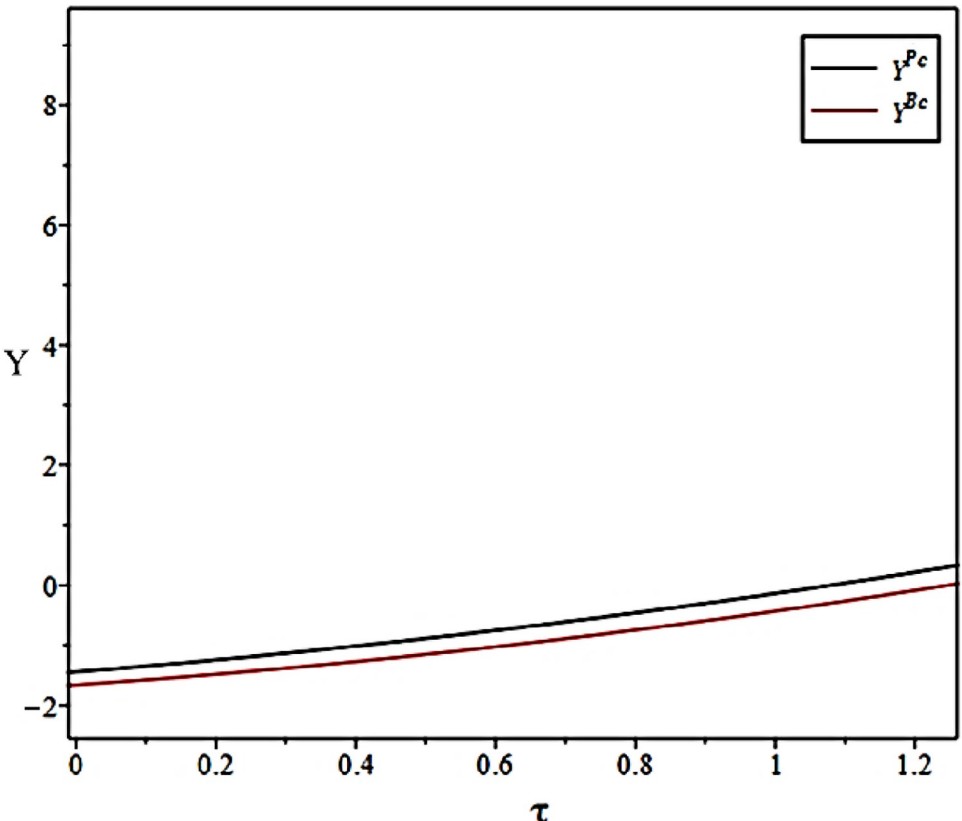

**Fig 10. Effect of synergy coefficients on preservation gains.**

preservation efforts of the CCL Service Provider and the incentive coefficients of the former are higher. (5) From the perspective of supply chain coordination strategy choices between the Fresh Food Producer and Distributor, adopting a collaborative approach does not immediately yield preservation benefits for both parties. However, as information exchange becomes more frequent and infrastructure is improved through joint efforts, the fresh food supply chain alliance can achieve higher long–term preservation benefits.

## 6.2 Future scope of research

There are some extensions suggested for future works. First, thus research could further investigate the decision–making and profit allocation in scenarios where multiple participants in the fresh agricultural product supply chain simultaneously outsource CCL services to the same provider. Instead of considering only two levels (producers and distributors), the study should take into account multi–level supply chains involving producers, distributors, and retailers. Second, while previous research focused on the principal–agent problem of cold chain services in the supply chain, it mainly discussed the distributor's role as the dominant participant. The situations where producers, distributors, and retailers each assume a dominant position in the supply chain and their decision–making processes for outsourcing cold chain services would be worth exploring. A Stackelberg game approach could be used for analysis in such cases. Third, the current model assumes risk–neutral participants in the fresh agricultural product supply chain and risk–averse behavior from the cold chain service provider. This assumption is idealized and may not reflect real–world scenarios accurately. In practice, the risk

preferences of both parties may change. Therefore, it would be interesting to research and analyze the principal–agent model with different risk preferences.

## Supporting information

**S1 Appendix.**
(DOCX)

**S1 Table.**
(DOCX)

**S1 Fig.**
(DOCX)

## Author Contributions

**Conceptualization:** Ming Zeng.

**Data curation:** Ming Zeng.

**Formal analysis:** Xiaoling Xing, Wenjing Tang.

**Funding acquisition:** Ming Zeng, Huyang Xu.

**Methodology:** Ming Zeng, Yuxiang Wu, Huyang Xu.

**Resources:** Ming Zeng, Yuxiang Wu, Huyang Xu.

**Supervision:** Xiaoling Xing, Wenjing Tang.

**Validation:** Ming Zeng.

**Visualization:** Ming Zeng, Xiaoling Xing, Wenjing Tang.

**Writing – original draft:** Ming Zeng, Yuxiang Wu, Xiaoling Xing, Wenjing Tang.

**Writing – review & editing:** Ming Zeng, Yuxiang Wu, Xiaoling Xing, Wenjing Tang.

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
