## [Decision Letter · Decision Letter 0]

30 Jan 2024

PONE-D-23-32982Coordinating principal-agent and incentive strategy of cold chain logistics service in fresh food supply chainPLOS ONE

Dear Dr. Wu,

Thank you for submitting your manuscript to PLOS ONE. After careful consideration, we feel that it has merit but does not fully meet PLOS ONE’s publication criteria as it currently stands. Therefore, we invite you to submit a revised version of the manuscript that addresses the points raised during the review process.

There are relevant issues that should be addressed. First, the paper's readability should be improved. Second, there are methodological aspects that need more work.

We look forward to receiving your revised manuscript.

Kind regards,

João Zambujal-Oliveira

Academic Editor

PLOS ONE

Journal Requirements:

3. We note that your Data Availability Statement is currently as follows: "All relevant data are within the manuscript and its Supporting Information files.".

Reviewers' comments:

Reviewer's Responses to Questions

**Comments to the Author**

1. Is the manuscript technically sound, and do the data support the conclusions?

Reviewer #1: Yes

Reviewer #2: Yes

2. Has the statistical analysis been performed appropriately and rigorously? 

Reviewer #1: Yes

Reviewer #2: No

3. Have the authors made all data underlying the findings in their manuscript fully available?

Reviewer #1: Yes

Reviewer #2: Yes

4. Is the manuscript presented in an intelligible fashion and written in standard English?

Reviewer #1: Yes

Reviewer #2: No

5. Review Comments to the Author

Reviewer #1: In this reviewed paper, the authors adopt coordination theory to research the delegation, coordination, and incentive strategies between fresh food producers, distributors, and cold chain logistics service providers. In order to deal with the complex problem, the basic models and a collaborative delegation agency model are established by the authors. Moreover, the authors discussed the motivational effect under different information conditions and collaborative cooperation strategies. I support its publication after considering the following comments.

1) For the studied problem, are there any other existing studies via coordination theory? If yes, please include them in your literature review. If possible, it is better to compare the performance of the suggested method and theirs.

2) It is missing an in-depth discussion considering this study’s findings and how they advance the literature.

3) Please highlight the contribution of this study, with reference to practice and contribution to the literature.

4) All figures should be improved for better reading.

5) English writing must be well-polished.

Reviewer #2: 1. The paper contains too many grammatical errors, spelling errors, punctuation errors, and formatting errors. Therefore, the authors are suggested to scan the entire text for possible English errors and rectify those accordingly. In addition, I think the writing of this paper could be improved in terms of logicality.

2. In Section 1, the authors are advised to provide answers to the research questions.

3. Why are Food Producer and Distributor both risk neutral while CCL Service Provider is risk averse? Please explain.

4. The authors actually set up the model based on the assumption in which costs except effect cost in the supply chain are standardized to 0. However, this assumption is not clearly demonstrated in Section 3, although the assumption is common in the existing literature (e.g., references [1] and [2]). Thus, the authors are advised to add the assumption and cite references [1] and [2].

5，Please provide reasons of each proposition in detail, rather than just paraphrasing the meaning of each proposition.

6. It is recommended that the Conclusion section should not exceed four paragraphs.

7. It is recommended that the authors provide future research in Section 6.

Overall, my recommendation is minor revision.

References

[1] "Manufacturer encroachment with a new product under network externalities." International Journal of Production Economics (2023): 108954.

[2] "Optimal mechanism for project splitting with time cost and asymmetric information." International Journal of Production Economics 264 (2023): 108987.

6. PLOS authors have the option to publish the peer review history of their article (what does this mean?). If published, this will include your full peer review and any attached files.

Reviewer #1: No

Reviewer #2: No

---

## [Author Response · Author response to Decision Letter 0]

12 Mar 2024

Cover Letter

Date: 15 March 2024.

Dear editor,

Thank you and the reviewers for commenting on our manuscript (PONE-D-23-32982) entitled “Coordinating principal-agent and incentive strategy of cold chain logistics service in fresh food supply chain”. We revised the manuscript in accordance with the reviewers’ comments, and highlighted the modified parts with a different color.

Reviewer 1

1. For the studied problem, are there any other existing studies via coordination theory? If yes, please include them in your literature review. If possible, it is better to compare the performance of the suggested method and theirs.

Reply: 

Thanks for the reviewer’s suggestion. Some representative studies via coordination theory have been added into the literature review of our revised manuscript. Also, the performance of the suggested method and the theirs have been compared comprehensively. For further details, please refer to the first and second paragraphs of section 2.3 Supply chain coordination and incentive strategies of the literature review.

2. It is missing an in-depth discussion considering this study’s findings and how they advance the literature.

Reply: 

The findings of our manuscript were explained thoroughly and an in-depth discussion about how they advance the literature were conducted. Please see the third paragraph of the final section 2.3 Supply chain coordination and incentive strategies of the literature review.

3. Please highlight the contribution of this study, with reference to practice and contribution to the literature.

Reply: 

The contributions of this study have been highlighted and clearly divided into theoretical and practical parts. The details can be found from the ninth paragraph of the introduction.

4. All figures should be improved for better reading.

Reply: 

All figures have been modified from the aspects of clarity, color, and line pattern for better reading. All images in our submitted revisions are attached separately.

5. English writing must be well-polished.

Reply: 

Thanks for this comment. The English expression of the full manuscript has been thoroughly checked and revised, all the changes were marked in red.

Reviewer 2

1. The paper contains too many grammatical errors, spelling errors, punctuation errors, and formatting errors. Therefore, the authors are suggested to scan the entire text for possible English errors and rectify those accordingly. In addition, I think the writing of this paper could be improved in terms of logicality.

Reply: 

Thanks a lot for the comments. The grammatical errors, spelling errors, punctuation errors, and formatting errors were carefully checked and the full text was examined in detail. Beyond, the logic of the writing was reorganized. 

2. In Section 1, the authors are advised to provide answers to the research questions.

Reply: Thank you for the suggestion. The answers of the three research questions were provided at the end of the Introduction Section.

3. Why are Food Producer and Distributor both risk neutral while CCL Service Provider is risk averse? Please explain.

Reply: The Food Producer and Distributor both risk neutral while CCL Service Provider is risk averse is based on the second hypothesis in our manuscript, which relies on the classic HM principal-agent theory elaborated in references Bernheim&Whitnston, 1985 and Mirrlees, 1976, as well as the traditional rational person hypothesis, which assumes that the principal, i.e. fresh agricultural product producers and distributors, is risk neutral and completely rational, while the agent, i.e. cold chain logistics service provider, is completely rational and risk averse. The above literatures have been added to the revised manuscript and explained in details. Please see Assumption 2 in Section 3.1.

4. The authors actually set up the model based on the assumption in which costs except effect cost in the supply chain are standardized to 0. However, this assumption is not clearly demonstrated in Section 3, although the assumption is common in the existing literature (e.g., references [1] and [2]). Thus, the authors are advised to add the assumption and cite references [1] and [2].

Reply: The hypothesis "costs except effect cost in the supply chain are standardized to 0" has been clearly demonstrated in Section 3 and the suggested references were cited in the revised version. The details can refer to Assumption 6 in Section 3.1. 

5. Please provide reasons of each proposition in detail, rather than just paraphrasing the meaning of each proposition.

Reply: 

The detailed information on how each proposition was derived has been supplied in the manuscript, please see XXX.

6. It is recommended that the Conclusion section should not exceed four paragraphs.

Reply: 

Thanks for the suggestion. The conclusion section has been reorganized and merge into three paragraphs. Please refer to Section 6.1.

7. It is recommended that the authors provide future research in Section 6.

Reply: 

The future scope of research has been extensively discussed in Section 6.2 in our revised manuscript. .

All the contents indicated above are in the revised manuscript. In summary, we appreciate the reviewers greatly for all the comments and suggestions, which are valuable in improving the quality of our manuscript.

Yours Sincerely,

Yuxiang Wu

College of Management Science

Chengdu University of Technology, China.

---

## [Decision Letter · Decision Letter 1]

27 Jun 2024

Coordinating principal-agent and incentive strategy of cold chain logistics service in fresh food supply chain

PONE-D-23-32982R1

Dear Dr. Wu,

We’re pleased to inform you that your manuscript has been judged scientifically suitable for publication and will be formally accepted for publication once it meets all outstanding technical requirements.

Kind regards,

João Zambujal-Oliveira

Academic Editor

PLOS ONE

Additional Editor Comments (optional):

Reviewers' comments:

Reviewer's Responses to Questions

**Comments to the Author**

1. If the authors have adequately addressed your comments raised in a previous round of review and you feel that this manuscript is now acceptable for publication, you may indicate that here to bypass the “Comments to the Author” section, enter your conflict of interest statement in the “Confidential to Editor” section, and submit your "Accept" recommendation.

Reviewer #2: All comments have been addressed

Reviewer #3: All comments have been addressed

2. Is the manuscript technically sound, and do the data support the conclusions?

Reviewer #2: (No Response)

Reviewer #3: Yes

3. Has the statistical analysis been performed appropriately and rigorously? 

Reviewer #2: (No Response)

Reviewer #3: Yes

4. Have the authors made all data underlying the findings in their manuscript fully available?

Reviewer #2: (No Response)

Reviewer #3: Yes

5. Is the manuscript presented in an intelligible fashion and written in standard English?

Reviewer #2: (No Response)

Reviewer #3: Yes

6. Review Comments to the Author

Reviewer #2: (No Response)

Reviewer #3: (No Response)

7. PLOS authors have the option to publish the peer review history of their article (what does this mean?). If published, this will include your full peer review and any attached files.

Reviewer #2: No

Reviewer #3: No

---

## [Editor Report · Acceptance letter]

30 Jul 2024

PONE-D-23-32982R1 

PLOS ONE

Dear Dr. Wu, 

I'm pleased to inform you that your manuscript has been deemed suitable for publication in PLOS ONE. Congratulations! Your manuscript is now being handed over to our production team.

Kind regards, 

on behalf of

Prof. João Zambujal-Oliveira 

Academic Editor

PLOS ONE